

# 1 Improving non-representative-sample prediction of forest
# 2 aboveground biomass maps: A combined machine
# 3 learning and spatial statistical approach

Shaoqing Dai [1,2], Xiaoman Zheng [1,2], Lei Gao [3], Shudi Zuo [1,2,4,] Qi Chen [5], Xiaohua
Wei [6], Yin Ren [1,4]
[1] Key Laboratory of Urban Environment and Health, Key Laboratory of Urban Metabolism of Xiamen,
Institute of Urban Environment, Chinese Academy of Sciences, CN 361021, China
[2] University of Chinese Academy of Sciences, CN 100049, China
[3] CSIRO, Waite Campus, Urrbrae, SA 5064, Australia
[4] Ningbo Urban Environment Observation and Research Station-NUEORS, Chinese Academy of
Sciences, CN 315800, China
[5] Department of Geography, University of Hawai'i at Mānoa, Honolulu, HI 96822, USA
[6] Department of Earth and Environmental Sciences, University of British Columbia, Kelowna, BC V1V
1V7, Canada
*Correspondence to*: Yin Ren(yren@iue.ac.cn)





**Abstract**:High-precision prediction of large-scale forest aboveground biomass (AGB) is important
but challenging on account of the uncertainty involved in the prediction process from various sources,
especially the uncertainty due to non-representative sample units. Usually caused by inadequate
sampling, non-representative sample units are common and can lead to geographic clusters of
localities. But they cannot fully capture complex and spatially heterogeneous patterns, in which
multiple environmental covariates (such as longitude, latitude, and forest structures) affect the spatial
distribution of AGB. To address this challenge, we propose herein a low-cost approach that combines
machine learning with spatial statistics to construct a regional AGB map from non-representative
sample units. The experimental results demonstrate that the combined methods can improve the
accuracy of AGB mapping in regions where only non-representative sample units are available. This
work provides a useful reference for AGB remote-sensing mapping and ecological modelling in
various regions of the world.

**Keywords:** Aboveground biomass map, Non-representative sample units, Machine learning, Spatial
statistical model, small-size samples



## 1 Introduction

Accurate mapping of aboveground biomass (AGB) can provide a precise scientific basis for decision-making in sustainable forest management, involving reducing deforestation, forest degradation, and greenhouse-gas emissions (Bustamante et al., 2016;Houghton et al., 2009;Mendoza-Ponce and Galicia, 2010). AGB maps are usually constructed based on biomass data obtained from small-size samples or geographically limited locations. The uncertainty in such maps can be attributed to two primary sources: (1) inadequate sampling designs used to obtain the data for constructing prediction models, especially geographically limited designs that do not capture the entire range of conditions; and (2) model-dependent uncertainty, including unreasonable model parameter assumptions as well as improper model structure(Chen et al., 2015;Gao et al., 2016;McRoberts et al., 2016).

An estimated 20%–50% of the uncertainty in AGB mapping can be attributed to the inadequate sampling design (Pelletier et al., 2011). To alleviate the uncertainty derived from inadequate sampling and its consequences (i.e., non-representative samples), one type of approaches focuses on processing model input samples (front-end processing), in the form of spatial filtering of existing sample units, quantification of sampling uncertainty, and acquisition of representative sample units (Boria et al., 2014;Galante et al., 2017;Marvin et al., 2014). Although the front-end processing approaches are widely used to reveal the distribution of biological populations, they are rarely used in AGB mapping because they make it difficult to quantify sampling uncertainty and produce large samples for spatial filtering methods. Another type of approaches, in the case of non-representative samples, increase the prediction accuracy by screening or building an optimal adaptive model (back-end processing) (Boria et al., 2014). These approaches may substantially increase the accuracy of AGB maps.

A sizable group of prediction models has been applied to constructing accurate AGB maps, including linear models (Andersen et al., 2014;Morel et al., 2012), machine learning models(Chen, 2015;Gleason and Im, 2012), and spatial statistical models (Benitez et al., 2016;Propastin, 2012;Van der Laan et al., 2014). With the development of computer-science techniques and advances in nonlinear biomass modeling, machine learning methods have become prevalent. Compared to traditional parametric methods (these methods summarize data with a fixed number of parameters with respect to the sample size, such as logistic regression and perceptron)(Gao and Hailu, 2012), which have difficulty in characterizing nonlinear relationships between AGB and multiple environmental covariates,





nonparametric machine learning algorithms (the number of parameters in a nonparametric method is
dependent of the number of training examples, e.g., K-nearest neighbor, support vector machine, and
random forest) are advantageous because they are more elastic and have neither restrictions on variable
types nor strict requirements regarding the distributions of predictor variables as well as the relationship
between response and predictor variables (Lu et al., 2007). In addition, nonparametric machine learning
algorithms may offer higher prediction accuracy (Frey et al., 2019;Gleason and Im, 2012).
Another frequently-used group of models for estimating relationships between forest AGB and multiple
environmental covariates is based on spatial statistical approaches, such as geographically weighted
regression and Kriging (Du et al., 2010;Van der Laan et al., 2014;Viana et al., 2012). Spatial statistical
methods are based on the analysis of attribute information that includes spatial locations (Schabenberger
and Gotway, 2005). Compared with traditional statistical methods, spatial methods integrate spatial
factors affecting model responses, thus remove the constraint of traditional statistical methods that
assume sample independence (Rangel and Bini, 2010) and improve the understanding of spatial
autocorrelation and heterogeneity (He et al., 2011;Rosenberg and Anderson, 2011).
Although many studies have integrated plot data, multi-source remote-sensing data (e.g., lidar and
Landsat), and machine learning or spatial statistical methods, the prediction accuracy of current AGB
spatial mapping still suffers from uncertainty (Asner et al., 2012;Chen et al., 2016;Gregoire et al.,
2016;McRoberts et al., 2018;Paul et al., 2016;Saatchi et al., 2011;Zheng et al., 2004) for two reasons.
First, the existing studies with machine learning methods do not consider the spatial heterogeneity of
multiple environmental covariates (such as longitude, latitude, and forest structures) that affect the spatial
distribution of AGB (Babcock et al., 2015;Fassnacht et al., 2014). Uncertainty can be further magnified
by applying regional area models to small-size samples or geographically limited samples. The second
reason lies in the assumptions of the spatial statistical method (e.g., spatial autocorrelation and stability
of the second steps), which may not always be valid in forest AGB.
The objective of this study is to develop and evaluate a method for improving the prediction accuracy of
large-scale AGB spatial mapping given small-size, non-representative, and local geographically
clustered samples. The method integrates the nonlinear mapping capabilities of machine learning
algorithms (artificial neural network, support vector machine, and random forest) with the spatial
autocorrelation and stratified heterogeneous advantages of a spatial statistical model ( the Point
Estimation Model of Biased Sentinel Hospitals-based Area Disease Estimation, P-BSHADE model)(Xu



et al., 2013). Our aim is tantamount to answer two specific questions: (1) What are the differences in
prediction accuracy of AGB maps for different machine learning methods and between machine learning
and spatial statistical methods? (2) Can the integration of spatial statistical and machine learning methods
improve the accuracy of AGB maps based on small-size, non-representative samples in the form of local
geographic clusters of forest inventory data? We explore these two questions by considering an empirical
case of predicting an AGB map for *Eucalyptus* plantation in Nanjing County, China.
**2 Materials and Methods**
**2.1 Site description**
Nanjing County (117°00'–117°36'E, 24°26'–25°00'N, Figure 1b) is located in the upper upstream area of
the Jiulong River, Fujian Province, China. Seventy-four percent (146,130 ha) of the county is covered
with forests, where 79,346 ha are plantations. The region is affected by the South Asian tropical monsoon
climate. In 2014, the average annual temperature in Nanjing County was 21.1°C, with an annual
precipitation of 1,700 mm and 340 frost-free days. Red soil is its major soil type.
The elevation in the study area varies significantly (0–1,566 m), with complex topography. There is also
major spatiotemporal heterogeneity in forest composition, structure, and biomass. The main types of
trees are *Eucalyptus*, *Pinus massoniana*, and *Cunninghamia lanceolata*. Recently the plantation area of
*Eucalyptus* has increased rapidly, reaching 13,305 ha and increasing by 10,862 ha in one decade.

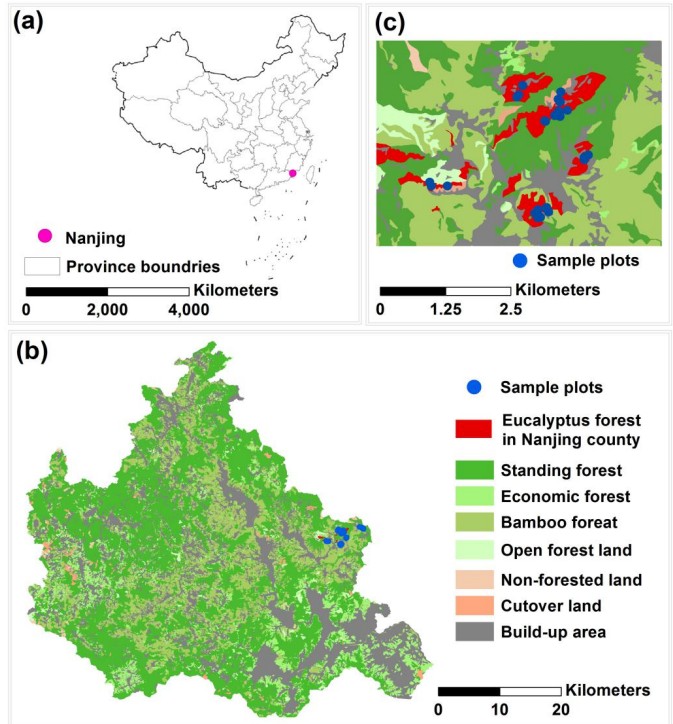

Figure 1. The study area is a typical example of a non-representative-sample problem. (a) Geographical location of the study area. (b) Spatial distributions of Eucalyptus plantations (red) and other major forests. (c) Spatial distributions of 30 sample plots (blue).

**2.2 Datasets**

The datasets included Forest Management and Planning Inventory (FMPI) data, sample plot data, and analytic-tree (destructive measurement) data.

**2.2.1 Forest Management and Planning Inventory (FMPI) data**

The FMPI data for the whole study area were provided by the Forestry Department of Fujian Province, China. By using large-scale sampling methods, this forest resource inventory collected detailed information about the characteristics and conditions of each type of forest. We selected the FMPI data of *Eucalyptus* plantation forest in this study.

The FMPI data were stored by patch and all trees with a diameter at breast height (DBH) greater than 8 cm were measured. The data contained (1) stand data (patch area, tree age which is the same for all trees in a given patch because they were planted at the same time, plantation density, mean DBH, mean tree



height, and total volume of each patch), (2) soil data (soil depth, humus depth, and site index of each
patch), and (3) topographical data (elevation, slope degree, slope direction, and slope position of each
patch). All variables were measured within each forest patch, with the average value being used as the
factor value for each patch. The accuracy of forest patch attributes was tested based on differences in
volume using a combined method of systematic and stratified samplings. A 95% sampling precision was
required. Table B.1 lists the statistical description of the forest patch data.
**2.2.2 Sample plot data**
A total of 30 fixed sample plots were selected in the Yongfeng forest farm. The plots were located in the
eastern part of the study area (Figure 1). The sample area accounted for 0.007% of the total area and
featured local geographic plot clusters. Thus, the sample size was small and the sample units were not
representative of the entire area. The 30 sampling plots with 10 age groups were built for *Eucalyptus*
plantation patches. In each plot, tree height (H) and DBH of each tree were measured. In addition, mean
plot-level variables were measured, including stand age, density, soil variables, and topographical
variables.
**2.2.3 Analytic-tree data**
The analytic-tree data were derived from standard wood in 30 fixed sample plots. Three trees were cut
in each of the plots, totaling 90 trees for 30 plots. We then calculated the biomass of each organ (foliage,
stems, and roots) for each tree. In addition, DBH and H were measured. Table B.2 presents the data of
90 parse trees. Details of the selection of standard wood and the cutting process are provided in S1 of
Supplementary Material.
**2.3 Construction of tree-level allometric models**
All analyses were based on the underlying assumption that the relationship between the response and
predictor variables in the sample data used to construct models was the same as the relationship in the
entire population. Using 90 analytic-tree data, three age groups (age 1-2, age 3-5, age 6-10) of allometric
models were constructed. Allometric models were then applied to each tree in each sample plot according
to their ages, hence producing a reference AGB of sample plots.



**2.4 Construct plot-level models to alleviate non-representative sample uncertainty**

The sample plots in this study were located in the east of the case study area and presented as non-representative samples (Figure 1). Processing based on model screening was applied to alleviate the uncertainty caused by non-representative samples and consisted of the following four steps (Figure 2).

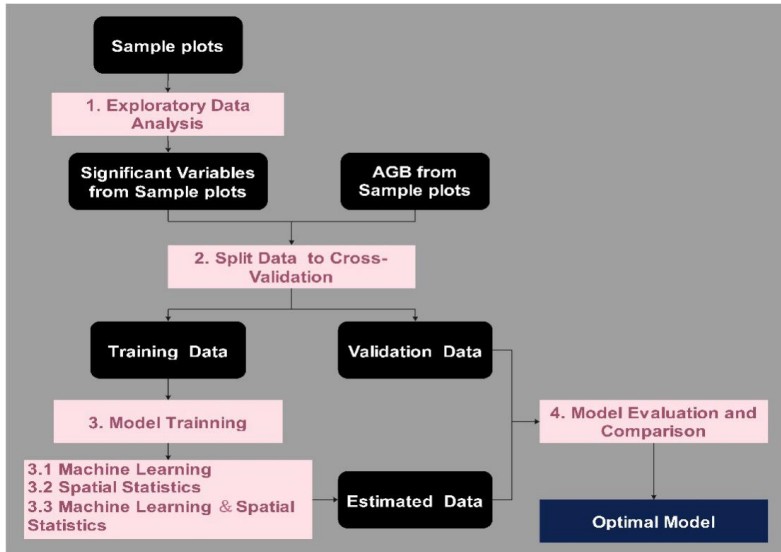

Figure 2. Structure of the optimal model screening scheme.

**2.4.1 Exploratory data analysis**

We first identified predictor variables used for creating the plot-level model. Based on our previous work (Ren et al., 2017), we selected plot-level environmental covariates including longitude and altitude, and forest attribute variables including forest distribution density, DBH, H, tree stem volume, timber volume, and forest age. The Pearson's correlation coefficient was used to investigate the correlation between these variables and the reference AGB of sample plots.

We then analyzed the spatial autocorrelation and spatial heterogeneity of AGB data from the selected non-representative sample plots. We used Moran's I (Cliff and Ord, 1981), a commonly used global spatial autocorrelation index, to evaluate spatial autocorrelation among the reference AGB of sample plots. The spatial stratified heterogeneity of the reference AGB of sample plots was evaluated using a geographic detector, as proposed by Wang et al. (2010).





### 2.4.2 Split data sets


We used the leave-one-out cross-validation method to split the 30 sample plots into 30 sets with each set
including two groups of data: validation data (one plot AGB) and training data (AGB and predictor
variables of another 29 plots), see Table B.3. The leave-one-out cross-validation method supposes that,
in an N-sample dataset, each sample is taken as a test sample, and the other N-1 samples are taken as
training samples. Thus, there are N iterations and we can obtain N datasets and N cross-validation results.

### 2.4.3 Model training


Seven models including three machine learning models (a, b, and c in Figure 3), one spatial statistical
model (d+e in Figure 3), and three combined machine learning and spatial statistical models (a+e, b+e,
and c+e in Figure 3) were developed and trained to simulate the reference AGB of sample plots (Figure
3). As shown in Figure 3, the three machine learning models are support vector machine (SVM, a), radial
basis function-artificial neural network (RBF-ANN, b), and random forest (RF, c) models. The spatial
statistical model, named P-BSHADE, required reference plot AGB data, which was obtained from the
localization biomass model (d). Thus, the single spatial statistical model (P-BSHADE, d+e) was
comprised of "d" combined with "e" in Figure 3. For the combined machine learning and spatial
statistical models, the reference plot AGB data in P-BSHADE was obtained from "a", "b" or "c". The
three combined models are represented as RBF-ANN&P-BSHADE (a+e), RF&P-BSHADE (b+e), and
SVM&P-BSHADE (c+e). Every model was trained based on each of 30 datasets, yielding a total of 30
simulated AGB datasets for 30 sample plots (see Table B.3).





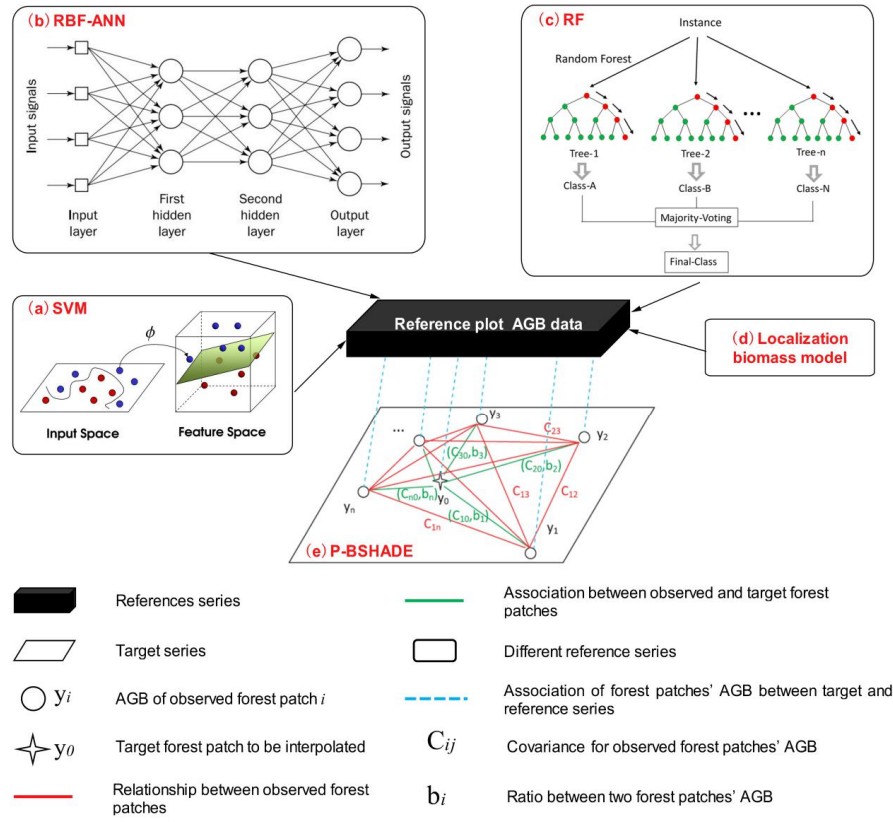


Figure 3. Framework for three machine learning (a, b, c), the PBSHADE (d+e), and three combined

machine learning and PBSHADE (a+e, b+e, c+e) models for AGB estimation.


(1) Machine learning

The SVM is a type of categorized algorithms that improves the generalized machine learning ability by

minimizing structural risks (so as to minimize the empirical risk and confidence intervals). In this way,

the SVM can achieve adequate statistical trends from a sample set of limited size(Drucker et al., 1996).

The basic components of the RBF-ANN include an input layer, a hidden layer, and an output layer, which

are able to provide the best approximation for nonlinear functions and optimal global performance

(Elanayar and Shin, 1994). The change from the input layer space to the hidden layer space is nonlinear,

whereas the spatial transformation from the hidden layer to output layer space is linear. The RBF network

not only has good generalizability, but also requires less calculation. In general, its learning speed is

faster than that of other machine learning algorithms, therefore, the lengthy process of iterative



calculations found in the learning algorithm of back propagation neural networks and the possibility of
falling into a local extremum can be avoided.
The RF is a relatively new machine learning technique. As one of modern classification and regression
methods, it is a combination of self-learning technologies (Breiman, 2001). The idea of combinatorial
learning is to integrate several individual classifiers when classifying new instances and to determine the
final classification of the instances by combining the classification results of multiple classifiers, so as to
achieve better performance than that achieved by each individual classifier.
The schematic function of machine learning is as follows
$$\mathrm{y}_j = f(x_{j,1}, x_{j,2}, x_{j,3}, x_{j,4}) \qquad (1)$$
where $\mathrm{y}_j$ is AGB of the $j$-th sample plot simulated by a machine learning model, $f(...)$ is a machine
learning model represented by a function of $x_{j,k}$ $(k = 1,...4)$, $x_{j,1}$, $x_{j,2}$, $x_{j,3}$, and $x_{j,4}$ are the
longitude, the DBH, the tree height, and the forest age of the $j$-th sample plot, respectively. A specific
description of the three machine learning models is given in S1 of Supplementary Material.
(2) Spatial statistical model: P-BSHADE
A spatial statistic model, P-BSHADE, was also used to estimate sample plot AGB. In essence, the P-
BSHADE uses the reference AGB of sample plots and the weights of target sample plots AGB against
reference AGB of each sample plot to obtain the AGB of the target sample plot. The P-BSHADE
assumption requires knowledge of the spatial autocorrelation and spatial stratified heterogeneity of the
reference AGB of sample plots. The specific mathematical expression of a P-BSHADE is as follows (Hu
et al., 2013;Xu et al., 2013):
$$\hat{y}_j = \Sigma_{i=1}^{n} w_{ij} y_i \qquad (2)$$
where $\hat{y}_j$ is the estimated AGB of the $j$-th sample plot by the P-BSHADE $(j = 1\sim30, n = 30)$, $y_i$ is
the reference AGB of the $i$th sample plot $(i = 1\sim30, n = 30)$, $w_{ij}$ is the weight (contribution) of
reference AGB of i-th sample plot to the AGB to be interpolated of $j$-th sample plot (when $j = 1, i =$
$2\sim30$; when $j = 1, i = 1$, $3\sim30$). A specific description of the P-BSHADE and the corresponding
algorithm formulas are presented in S1 of Supplementary Material.
(3) Combination of machine learning and spatial statistical models
P-BSHADE was separately integrated with three machine learning methods (SVM, RBF-ANN, and RF)
to form three combined models (SVM&P-BSHADE, RBF-ANN&P-BSHADE, and RF&P-BSHADE).
The reference AGB of 30 sample plots were replaced by the estimates produced from machine learning



models. A combined model can be represented as follows
$\hat{y}_j = \Sigma_{i=1}^n w_{ij} y_i$ (3)
where $\hat{y}_j$ is the estimated AGB of the $j$-th sample plot using the combined model $(j = 1{\sim}30, n = 30)$,
$y_i$ is AGB estimated by machine learning based on the i-th sample plot $(i = 1{\sim}30, n = 30)$, $w_{ij}$ is
the weight (contribution) of $i$th machine learning estimation AGB of the sample plot to $j$-th sample plot
AGB to be interpolated (when $j = 1, i = 2{\sim}30$; when $j = 1, i = 1, \ 3{\sim}30$). A specific description of
the combined models and the algorithm formulas are presented in S1 of Supplementary Material.
**2.4.4 Model evaluation and comparison**
To evaluate the prediction performance of the seven models (SVM, RBF-ANN, RF, P-BSHADE,
SVM&P-BSHADE, RBF-ANN&P-BSHADE, and RF&P-BSHADE), the AGB results simulated by the
seven models were compared to the reference AGB of sample plot groups (AGB group M in Table B.3)
in terms of three performance indicators: mean absolute error (MAE), mean relative error (MRE), and
root mean square error (RMSE), as shown in Eq. (4)-(6).
$\text{MAE} = \left(\sum_{i=1}^n |y_i^p - y_i|\right)/n$ (4)
$\text{MRE} = \left(\sum_{i=1}^n |y_i^p - y_i|\right)/(y_i \times n)$ (5)
$\text{RMSE} = \sqrt{\left(\sum_{i=1}^n (y_i^p - y_i)^2\right)/n}$ (6)
where $y_t^p$ is the predictive value of the different models, $y_i$ is the AGB of the $i$th sample plot, and $n$
is the number of training datasets.
Then, in terms of the calculated MAE, MRE, and RMSE, we identified the optimal model.
**2.5 Model application**
We applied the optimal model to each *Eucalyptus* forest patch and estimated the total AGB over all
patches in the study area. In short, the relationship between the non-representative AGB data from the
sample plots and their covariates were applied to each *Eucalyptus* forest patch in regional forests to
estimate the AGB of the area.
To validate the estimated AGB map, we compared it with the AGB map obtained by an allometric model,
and 95% credible interval width (CIW) was calculated and mapped for AGB. The allometric model was
expressed as the formula $AGB = a(D^2H)^b$, where D is the breast height (m), H is the tree height (m),



and a and b are constants. This model is acknowledged as a fast, simple, and basic method to calculate
regional AGB. In our study, we used the AGB, mean H, and mean D of 30 sample plots to constitute the
allometric model.
**3 Results**
**3.1 Reference AGB of sample plots**
The range of reference AGB of these 30 sample plots was calculated as 1.02~135.79 Mg·ha$^{-1}$·plot$^{-1}$,
with an average value of 47.34 Mg·ha$^{-1}$·plot$^{-1}$ and a standard deviation of 34.46 Mg·ha$^{-1}$ plot$^{-1}$. The
coefficients of variation of the AGB for all the sample plots and for the 10 age categories were calculated
as 0.73 and 0.07~0.37, respectively.
**3.2 Exploratory data analysis**
**3.2.1 Selection of variables**
Figure 4 shows the correlation-coefficient matrix of variables. The following variables are strongly
correlated with AGB: longitude $(r = -0.56)$, diameter at breast height $(r = 0.79)$, tree height
$(r = 0.84)$, trunk volume $(r = 0.86)$, timber volume $(r = 0.98)$, and forest age $(r = 0.82)$. The AGB
map for the *Eucalyptus* forest in Nanjing is based on the data from the forest resource inventory; therefore,
the selected covariates should be accessible from the forest resource inventory dataset. Because the
timber volume and stem volume were both estimated based on tree height and diameter at breast height,
they were excluded as covariates for the AGB mapping. To summarize, four variables (longitude,
diameter at breast height, tree height, and forest age) were selected as covariates for the AGB mapping
of the *Eucalyptus* forest in the Nanjing region. Table B.4 lists the statistical descriptions of these
covariates and the AGB statistics for the 30 sample plots.





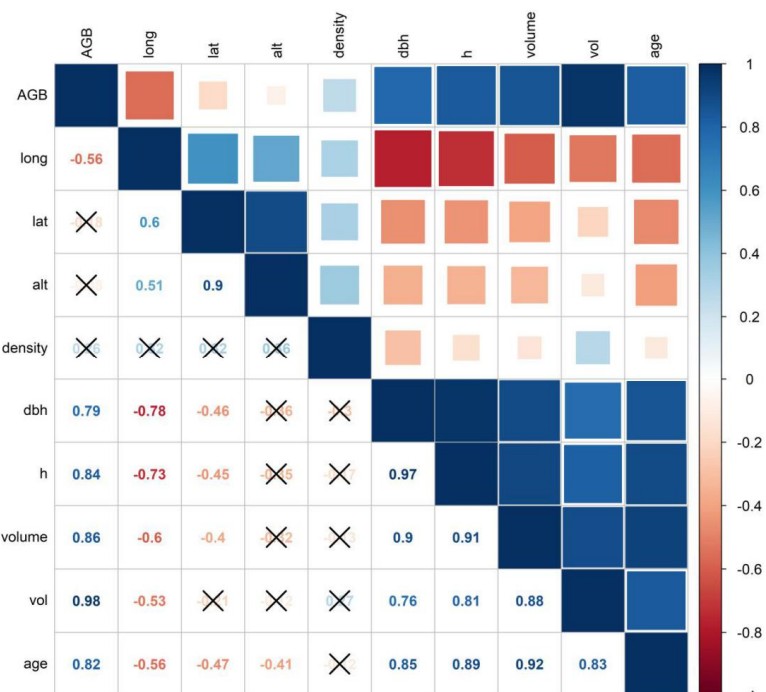


Figure 4. Pearson's correlation coefficients between AGB and other variables represented by numbers

and squares. Negative numbers are negatively correlated and are colored in red, while positive blue

numbers represent positive correlations. Larger absolute numbers, darker colors, and larger squares all

indicate stronger correlation, while $\times$ indicates the variables were uncorrelated.

**3.2.2 Spatial autocorrelation test**

The spatial distribution of the reference AGB of the 30 sample plots shows a pattern of aggregation (see

red part in Figure C.1 in the supplementary material and Table 1). In addition, because less than 1% of

the AGB data is randomly distributed (see blue part in Figure C.1 and Table 1), the possibility of

aggregation distribution is greater than that of random distribution. Furthermore, the null hypothesis is

significantly rejected ($p < 0.01$). These results show that the spatial distribution of the AGB data displays

aggregation and a pattern of strong spatial autocorrelation.






Table 1. Spatial autocorrelation and heterogeneity test

| Spatial autocorrelation | | Spatial heterogeneity | | |
|---|---|---|---|---|
| **Items** | **Values** | **Factors** | **q-value** | ***p*-value** |
| Moran I | 0.36 | AGB | 0.87 | <0.01 |
| | | Longitude, long | 0.38 | <0.01 |
| z-score | 4.78 | Diameter at breast height, dbh | 0.54 | <0.01 |
| | | Tree height, h | 0.63 | <0.01 |
| *p*-value | 0.00 | Age | 0.92 | <0.01 |

**3.2.3 Spatial heterogeneity test**
As shown in Table 1, the reference AGB of sample plots can be divided into three strata using $K$ means
clustering with a $q$ value of 0.87 and a $p$ value less than 0.01. These results indicate that the within-
layer variance is far less than the sum of variances among different strata. The results also show that the
reference AGB of 30 sample plots is associated with obvious spatial differentiation.
**3.3 Performance of models**
We developed seven models for AGB estimation: three machine learning models (SVM, RBF-ANN, and
RF), one spatial statistical model (P-BSHADE), and three combined models that integrated each machine
learning method with the spatial statistical method (SVM & P-BSHADE, RBF-ANN & P-BSHADE, and
RF & P-BSHADE). Furthermore, we used the leave-one-out cross-validation method to split the datasets
and evaluated the prediction performance of these seven methods in terms of the indicators of MAE
(Figure 5a), MRE (Figure 5b) and RMSE (Figure 5c).





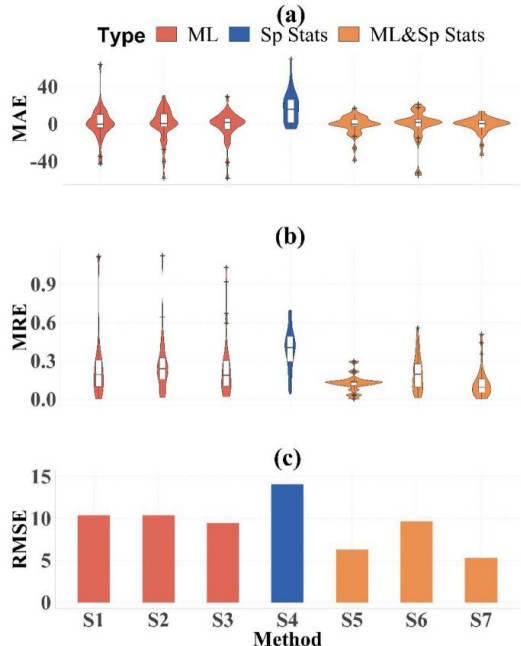


Figure 5. Results of prediction performance of the seven different models. The MAE (a) and MRE (b)
presented by boxplots for each prediction method (S1=SVM, S2=RBF-ANN , S3=RF, S4=P-BSHDE ,
S5=SVM & P-BSHDE, S6=RBF-ANN & P-BSHDE, S7=RF & P-BSHDE, ML=machine learning, Sp
Stats=Spatial statistic), with the median (black line in the box), inter-quartile range (25%-75% in the
box), the range 5%-95% (whiskers), and outliers (asteroids) labeled. The histogram distributions of
RMSE for each prediction method are presented in Figure 5 (c).

Compared with the calculated indicators by the P-BSHADE (MAE=18.37 Mg·ha$^{-1}$, MRE=39.13%, and
RMSE=14.08 Mg·ha$^{-1}$), the forest AGB estimate obtained by the three machine learning methods has a
MAE of 10.16~12.15 Mg·ha$^{-1}$, a MRE of 24.79~26.69%, and a RMSE of 9.43~10.39 Mg·ha$^{-1}$, which
are substantially smaller than those obtained by the spatial statistical method.
Among the three machine learning methods, the accuracy of RF is the highest and its three evaluation
indexes are MAE=10.16 Mg·ha$^{-1}$, MRE=25.93%, and RMSE=9.43 Mg·ha$^{-1}$, which are not only
substantially smaller than those for P-BSHADE, with MAE=18.37 Mg·ha$^{-1}$, MRE=39.13%, and
RMSE=14.08 Mg·ha$^{-1}$, but are also smaller than most of those obtained by the other two machine
learning methods with MAE=11.17~12.15 Mg·ha$^{-1}$, MRE=24.79~26.69%, and RMSE=10.39~10.39





Mg·ha$^{-1}$.
Finally, compared with single machine learning methods, the combination of machine learning and
spatial statistical models produced smaller MAE (5.68~10.14 Mg·ha$^{-1}$), MRE (12.47~20.49%), and
RMSE (5.30~9.63 Mg·ha$^{-1}$). In addition, among the three combined methods, the combination of random
forest and the spatial statistical model (RF&P-BSHADE) produced a higher accuracy with the smallest
MAE (5.68 Mg·ha$^{-1}$), modest MRE (12.97%), and smallest RMSE (5.30 Mg·ha$^{-1}$). In contrast, the MAE
(10.14 Mg·ha$^{-1}$), MRE (20.49%), and RMSE (9.63 Mg·ha$^{-1}$) of RBF-ANN&P-BSHADE were the
highest among the three combined methods. Furthermore, compared with the RBF-ANN&P-BSHADE
model, the RF&P-BSHADE model achieved a reduction of the cross-validated prediction error of 36.73–
44.99% (43.97% for MAE, 36.73% for MRE, and 44.99% for RMSE).
**3.4 Model application and mapping of AGB**
Figure 6(a) shows the spatial distribution of the AGB predicted by the RF&P-BSHADE. The AGB
simulated by RF&P-BSHADE is 7.54~89.93 Mg·ha$^{-1}$, with an average of 41.21 Mg·ha$^{-1}$, a median of
43.53 Mg·ha$^{-1}$, a standard deviation of 18.83 Mg·ha$^{-1}$, and a coefficient of variation of 45.69%. The 95%
predictive distribution credible interval width (CIW) was calculated and is mapped for AGB in Figure
6(b). Wide CIWs are distributed not only in the high-altitude areas, but also in the low-altitude areas
which are easier to access.

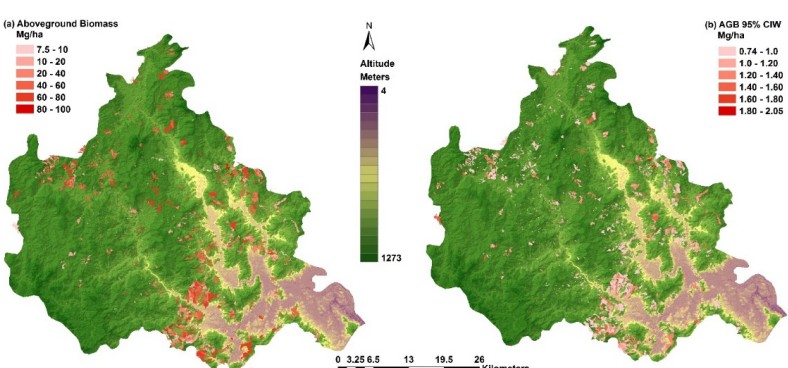


Figure 6. Map of AGB (a) and associated 95% credible interval width (CIW) (b) using RF&P-

BSHADE. This map shows two main areas: (1) red, study area of *Eucalyptus* plantations, and (2)

green, outside of study area.




The total AGB of the Nanjing area (2,980 forest patches) estimated by RF&P-BSHADE is 122,812.1
Mg, and that estimated by the allometric model is 123,021.5 Mg. The relative percent difference in total
AGB between the two methods is 0.17%.
**4 Discussion**
**4.1 The significance of the AGB map at the regional scale**
In the past, ecologists would often assume that a limited number of sample plots could be used to
represent a large range of landscapes, and such sample plots have long served as the main source of
information for understanding the spatial distribution of AGB from the sample-plot scale to the regional
scale. However, the research by Marvin et al. (2014) confirmed that the distribution of most AGB is non-
Gaussian, skewed, or multi-modal, especially in tropical and subtropical regions. Marvin et al. (2014)
asserted that the most influential source of uncertainty is the non-representativeness in the sample design
in the form of local geographic clusters of sample units. Therefore, AGB maps based on non-
representative samples introduce greater uncertainty. For example, in the Amazon basin, fewer than
500 geographically concentrated sample plots were used to represent more than 109 hectares of forest,
thus undoubtedly contributing to relatively large uncertainty (Mitchard et al., 2014). However,
reducing the uncertainty in AGB maps to levels corresponding to high precision would require unrealistic
sample sizes; for example, 44 low-lying 1 ha sample plots or more than 85 mountain 1 ha plots are
required for every 100 ha on an AGB map (Mitchard et al., 2014). Inevitably, the area represented by
these geographically concentrated plots is much less than the total area of the tropical forest represented
in the final map. Provided that the limited sample size cannot represent the spatial heterogeneity of the
large-scale area, subsequently, the AGB map cannot lead to reliable quantitative conclusions (Duncanson
et al., 2015).
To overcome the small sample size and non-representative sample problems which lead to
geographically concentrated local plot clusters, we integrated the advantages of machine learning and
spatial statistics at a regional scale (the key region linking the sample plots to the landscape scale) to
construct an AGB map for a subtropical region. The approach provides not only a low-cost, high-
precision map of AGB whose estimates can be compared with those obtained from remote sensing,





ground observation, and model simulation, but also a scientific basis to assist forest-management
decisions (e.g., the quantitative evaluation of carbon emissions from deforestation). Combining the
advantages of machine-learning-based quantification of AGB and the complex nonlinear relationship
between multiple environmental covariates, in conjunction with the proposed P-BSHADE model, the
spatial correlation and heterogeneity of multiple environmental covariates are incorporated into the
model, and the sample points are subsequently rectified, thus leading to the best linear unbiased estimate
(BLUE) of the target site. Given that current multi-source databases cannot provide high-precision
accuracy of mapping affected by the variations of AGB in subtropical areas, especially in regions with
large variability, current studies mainly use fusion maps composed of different and independent data sets
(Avitabile et al., 2015). Therefore, we provide the most accurate AGB map by data fusion of single
analytic trees and forest resource inventory data which may be used to extrapolate AGB from the tree
scale to the field and regional scales.

**4.2 Benefits of random forest in predicting an AGB map**

This study shows that among the three machine learning methods, the prediction accuracy of random
forest in AGB mapping is the highest. This is consistent with the results from Gleason and Im (2012)
and Fassnacht et al. (2014). For example, Fassnacht et al. (2014) combined lidar with multiple remote-
sensing data, such as airborne hyperspectral data from Karlsruhe, Germany, to compare the AGB
prediction accuracy of five machine learning methods: stepwise regression, support vector machine,
random forest, Gaussian processes, and K-nearest neighbor. The evaluation indexes for leave-one-out
cross validation (i.e., $R^2$ and RMSE) showed that the random forest method was associated with the
highest prediction accuracy due to self-learning techniques of the random forest method. The random
forest method clearly differs from the other machine learning methods in the flexibility of its conceptual
design and method. In detail, the following advantages of random forest method may help improve the
precision of predicting an AGB map (Breiman, 2001): (1) The random forest method can generate highly
accurate classifiers, detect the interaction between variables, and also detect outliers and monitor data;
(2) For unbalanced and categorized data sets, the random forest method can balance the deviations; (3)
The random forest method can be extended to unlabeled data, which usually use unsupervised clustering;
(4) In the construction of a forest, the random forest method can internally produce unbiased estimates
for generalized deviations; (5) The random forest method contains a good way to estimate missing data.



In addition, if a large part of the data is missing, the random forest method can still maintain accuracy.

**4.3 Machine learning outperforms the spatial statistical model in prediction performance**

Regarding the AGB mapping of non-representative sample units, the machine learning methods
outperformed the spatial statistical method (P-BSHADE) in the prediction accuracy. This may be because
machine learning offers an array of supervised learning models capable of relating forest AGB to multi-
variables including forest variables and environmental variables via complex, potentially nonlinear
functional relationships. Machine learning models appear to be good at tackling high-dimensional
problems, particularly in areas where a lack of knowledge exists regarding the development of effective
algorithms, and where programs must dynamically adapt to changing conditions (Görgens et al.,
2015;Latifi et al., 2010;Stojanova et al., 2010). In addition, the P-BSHADE model yielded negative
weights between a small number of patches which might introduce slight uncertainty into the result (Xu
et al., 2013). Our results were consistent with the study of Povak et al. (2014) and Li et al. (2011), who
found that a machine learning method (RF) outperformed the spatial statistical method (e.g.,
Geographically Weighted Regression, Inverse Distance Weighting ) in terms of prediction accuracy.

**4.4 Why a combined model outperforms a single machine learning or spatial statistical model**

As expected, the prediction accuracy of the combined methods is higher than that of any single method
(either a machine learning or a spatial statistical). In the previous sections, we described how the
advantages of the P-BSHADE model can compensate for the inherent defects of machine learning.
Virtually, the P-BSHADE model is also handicapped by the fact that the founding assumption does not
conform to reality. The assumption is that the AGB is accurate in all other sampling plots except at this
target sampling plot. In reality, each sampling plot has a varying degree of AGB uncertainty. In other
words, the premise behind only using the P-BSHADE model is that the reference AGB data is accurate.
Since the P-BSHADE model combined with machine learning uses the results optimized by machine-
learning as the reference values; therefore, it further improves the accuracy of AGB mapping. Machine
learning methods or the P-BSHADE model have been adopted to model the uncertainty of temperature
observation obtained by weather stations (Fassnacht et al., 2014;Paul et al., 2016;Xu et al., 2013).
However, methods in these studies were adopted independently. Conversely, the combination of machine
learning and spatial statistics can improve the prediction accuracy of AGB maps, which in turn can be



used as criteria for improving the accuracy of lidar remote-sensing technology and the results of
ecological-process models. Eventually, these achievements can promote process-oriented projects of
dynamic AGB predictions for large-scale forests in different forest-management scenarios.
In addition, we compared the prediction accuracy of AGB mapping obtained by the combined spatial
statistical and machine learning models with that reported by recent local and international research into
AGB mapping. In the current literature on remote-sensing estimation of forest AGB, RMSE and $R^2$ were
commonly used as indexes for evaluating prediction performance when these studies looked at the
importance of research sample size, data types, and forecasting methods (Fassnacht et al., 2014). In
contrast, our study uses three conventional indexes for evaluating prediction performance: RMSE, MAE,
and MRE. Because the main goal of this work is to predict regional forest AGB based on a small number
of non-representative sample units, the criterion of model selection is to choose indexes summarized
from sample prediction (such as RMSE), rather than choosing the goodness-of-fit $R^2$ (Babcock et al.,
2015). Based on calculated RMSE indexes, the AGB prediction accuracy of the combined random forest
and P-BSHADE method (5.30 Mg·ha$^{-1}$) is higher than that obtained by Babcock et al. (2015)(34.21
Mg·ha$^{-1}$) in Colorado, USA, where the authors used a combination of airborne lidar, forest inventory
database, and a Bayesian spatial hierarchical framework model and introduced spatial random effects to
compensate for the residual spatial dependence and nonstationarity of model covariates. In addition,
prediction accuracy of AGB in this work is also higher than that obtained by (Ene et al.,
2016)(RMSE=15.92 Mg·ha$^{-1}$) in southeast Norway using a general linear regression model with airborne
lidar and ground survey. Furthermore, the prediction accuracy of AGB in this work is also higher than
those obtained by (Avitabile et al., 2015) in the tropics (Central America: 22.8±0.3 Mg·ha$^{-1}$; Africa:
83.7±2.5 Mg·ha$^{-1}$) using fusion maps of multi-source databases combined with the random forest method.
Our prediction performance is close to that obtained by Marvin et al. (2014) (6 Mg C·ha$^{-1}$) who studied
the Amazon tropical forests using a Monte Carlo method based on airborne lidar in conjunction with on-
site monitoring. Because RMSE is an absolute measure of the deviation between the predicted and the
observed data, a large range of reference values may cause large deviations. With our combined methods,
the calculated RMSE for the prediction accuracy of AGB is relatively small, which we attribute to the
following reasons: (1) The reference AGB of 30 sample plots were calculated from each tree by the
allometric model constructed with 90 most accurate analytic trees. There were no differences in the range
of reference values. (2) Machine learning methods were used to quantify the complex nonlinear



relationship between AGB and multiple environmental covariates. (3) We applied a statistical method
based on the hypothesis of spatial heterogeneity. Although the RMSE index was calculated by different
studies using different datasets and prediction methods in different locations, most studies deemed that
RMSE was the most commonly used indicator for measuring the prediction errors of remote-sensing
AGB models and calculating the real AGB of forest sample plots. In contrast to other studies, our work
reflects not only our attention to subtropical forests, but also the methodological differences in
uncertainty mitigation, especially in comprehensively addressing the sources of uncertainty caused by
multiple spatial and environmental covariates.
**4.5 Comparison of RF&P-BSHADE with the allometric growth model**
Because the allometric growth model can offer a fast and simple calculation method, it has been used as
the basis for determining the benchmark map in quite a few studies. Nevertheless, spatial heterogeneity
caused by multiple environmental covariates is not considered in the allometric model, as there may be
errors in the AGB estimate and the errors may be propagated to affect the accuracy of the regional AGB
benchmark map. This study shows that the relative percent difference in total AGB between RF&P-
BSHADE and the allometric method is 0.17%. Meanwhile, the MRE of AGB between the two methods
ranged from 0.04% to 99.8% with an average of 19.93%. These results confirm that the RF&P-BSHADE
estimates can be used as the main reference for regional-scale forest AGB maps. Furthermore, it also
shows that the two methods are roughly the same in terms of overall estimates of AGB, but the local
spatial distribution of AGB is different. The differences in AGB spatial distribution have been reported
in many studies of AGB maps. Babcock et al. (2015) asserted that the main reasons for the differences
in the spatial distribution of AGB maps between different methods include the following: (1) The
structural framework of different research methods and schemes cannot truly reflect the actual situation
of the forest growth. (2) The model is usually a simplification of an ecological process and ignores the
spatial heterogeneity at the regional scale. (3) The model does not consider the influence of multiple
environmental covariates (vegetation, topography, etc.) on forest growth in the region.
**4.6 Implications for AGB mapping and future research directions**
Based on the results of this study, we have the following two implications. First, to enhance the prediction
accuracy of large-scale AGB mapping, we should not only reduce the effect of sampling uncertainty by





improving the sampling method (by data treatment such as quantification of sampling errors and spatial
filtering of existing data sets), but also solve the problems of nonlinearity, complexity, and spatial
heterogeneity from the perspective of both model and algorithm. Second, in all probability, the sampling
plots for the real values on the ground are only accessible in small sampling areas within non-
representative locations. Therefore, the combined use of spatial-differentiation-based statistical analysis
and machine learning with nonlinear fitting should improve the prediction accuracy of AGB mapping.
Additionally, more machine learning methods (such as KNN algorithms) can be tried and combined with
P-BSHADE in future research to explore the best AGB mapping methods for large-scale forests. The
case we present herein is only for a pure *Eucalyptus* forest, and further research can create separate
databases for different forest types in a complex tropical forest system to create a hierarchical mapping.
If the identification of plant species is also included in field plot-based AGB assessment and monitoring,
such identification information can also provide important information about changes in species
composition. Overall, forest AGB mapping should not be static. Instead, it should be generated based on
time sequences using an ecological-process model, so as to capture the changes in the AGB map database
over time(Bustamante et al., 2016). In addition, more environmental and socio-economic datasets (for
example, the meteorological variables that are missing in the present study) should be included and the
correlation between them should be taken into account in the future work.
**5 Conclusion**
Currently, extrapolations and predictions based on sparse and/or non-randomly distributed forest plots
cannot solve the problem of regional carbon balance in tropical forests. With the continuous development
of remote sensing, ground observation, and methods of ecological-process modeling, the number of
global and regional AGB datasets is continuously increasing. As criteria to judge the differences between
different estimates of biomass, an AGB map not only provides a decision-making basis for forest
managers to mitigate the negative impact of climate change, but also helps different countries evaluate
and implement the policies and programs that aim at reducing regional-scale deforestation and forest
degradation, so as to avoid more carbon emissions.
Given the conditions of insufficient sample size and non-representative sample units that lead to
geographic clusters of localities, we propose a method to integrate the advantages of machine learning





and spatial statistics, different datasets, and multiple environmental covariates, to solve the problem of
uncertainty in regional AGB maps. Based on the most accurate data for single analytic trees and forest
resource inventory data, we extrapolate the study from the single-tree to the regional scale. In this study,
although the forest resource inventory data and the data of analytic-trees are solely available for
*Eucalyptus* forests located in the Nanjing area of China, the proposed method and the findings can
provide references for AGB remote sensing and simulation of ecological processes in different countries
and in different types of tropical forests.
**Acknowledgements**
Shaoqing Dai and Xiaoman Zheng contributed equally to this work and should be considered as co-lead
authors. This work was supported by National Science Foundation of China (31670645 and 31470578),
the National Key Research Program of China (2016YFC0502704), National Social Science Fund
(17ZDA058), Fujian Provincial Department of S&T Project (2016T3032, 2016T3037, 2016Y0083,
2018T3018), Key Laboratory of Urban Environment and Health of CAS (KLUEH-C-201701) and Key
Program of the Chinese Academy of Sciences (KFZDSW-324). We are grateful to Professor Li Hu for
his helpful suggestions.

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
