# Peer review of "Improving non-representative-sample prediction of forest"

_Biogeosciences, 2019_

## Referee Comment (RC1) · Anonymous Referee #1 · 27 Jun 2019

The study addresses the problem that maps of forest aboveground biomass (AGB) are often based on sample plots which may not be representative for the entire area of interest. In a case study for Eucalyptus plantations in southern China, the authors compare AGB predictions produced with different approaches. These include three machine learning methods and one spatial statistical method as well as combinations of them. Based on a set of goodness-of-fit statistics, the authors conclude that a combination of random forest and the spatial statistical approach P-BSHADE produce the best AGB predictions and that this approach could reduce uncertainty due to nonrepresentative sampling. Non-representative sample distribution is indeed a big problem in mapping of AGB and other forest attributes. However, the manuscript does not clearly state how the presented methodology can improve this situation. While research question (1) can partly be answered by the given results (e.g., Fig. 5), a clear answer to research question (2) is not possible based on the conducted analyses. Several sections need considerable improvements for the manuscript to become a consistent story.

In the following, I present detailed comments:

Abstract:

The Abstract currently focusses mostly on the problem statement. It should be more specific about the used methods and most important results.

Introduction:

In general it should be made clear that this study does not use remote sensing data for AGB mapping. As many studies do use remote sensing and the authors cite several such studies, it is not clear to the reader that the forest structure used in this study is entirely ground based. This should be clear from the beginning.

L. 53: Please explain what is meant by back-end processing. The difference to front-end processing is not clear.

L. 86: Explain what is meant with stability of the second steps.

Materials & Methods:

L. 111: According to the numbers given in L. 103 and 110 about 10% of the forest in Nanjiing county is Eucalyptus forest. So why isn't ~10% of Fig. 1b red? Also mind the typo "Bamboo foreat" in legend.

L. 119: Can you describe the FMPI data a bit further? Are these 2980 patches polygons of irregular shape representing stands of same structure? If yes, what are typical

sizes? Or are they grid cells?

L. 126: Please remove information irrelevant in this study. I think, soil and topographical variables are not further used in the analysis.

L. 129-131: It is unclear what is meant by systematic and stratified sampling in this context.

L. 142: Foliage and roots should not be mentioned if they are not relevant for the analysis.

L. 156: Typo in Fig. 2 "Model Trainning".

L. 167: Please explain briefly the concept of spatial stratified heterogeneity and how the geographic detector method works.

L. 178: At several places in the manuscript the word "simulate" is used, where I would use "predict". Simulation is usually associated with process-based model simulations, while statistical model output is more commonly referred to as prediction. Consider changing the wording to avoid confusion.

L. 182: What do you mean with "localization biomass model"? To my understanding the P-BSHADE interpolates the AGB values of the neighboring plots.

L. 192-208: The descriptions of all machine learning methods are hard to understand. I suggest to simplify and only explain the principles of each of them. Technical details are confusing if they are not explained at length. Since all are well known machine learning methods, I would refer to the original literature for details.

L. 213: Is it mean DBH and mean tree height?

L. 215-227: The explanation of P-BSHADE is not sufficient to understand the method. Since this is not a standard method it disserves more explanation. Again, for technical details I would refer to the original literature, but the principle ideas should be clearly explained. Also, how is it related to existing methods like kriging and inverse distance

weighted interpolation? What are the differences to them? Does distance between plots play a role and how many neighboring plots are considered for a prediction?

L. 252: The number of patches (2980) should be mentioned here to make clearer that you are not speaking of the 30 plots here.

L. 258: D is probably not breast height, but diameter at breast height.

L. 256-261: This is not a validation! This is a comparison of AGB predicted with the optimal model to AGB predicted via a simple allometry. A validation would imply that the data used for comparison is (a close approximation of) the truth. Then it could be tested how close the predictions can get to the truth. But in the case here, the authors are not stating that they consider the simple allometric AGB to be the truth. It is not clear why they make this comparison and also not the role of the CIW.

Results:

L. 264: Change Mg ha-1 plot-1 to Mg ha-1.

L. 280: Please use clear variable names in Fig. 4. E.g., "vol" and "volume" are ambiguous.

L. 296-300: To understand this a more detailed explanation of the method is required (see comment on L. 167). The k-means clustering was not mentioned before and a reader unfamiliar with the method does not know how to interpret q-values.

L. 308: This graphic nicely summarizes the main results. However, it raises two crucial questions, which are not answered in the text:

1) If S4 are interpolations based on the actual AGB values of the other plots and S5 to S7 are interpolations based on ML-predicted AGB of the other plots, how can S4 be worse than S5 to S7? The ML-predictions should introduce additional uncertainty compared to S4 which uses the actual AGBs. This has to be explained.

2) If we accept the fact questioned in my question 1), it remains the question why the

combined approach can strongly improve the SVM and RF approaches, but not the ANN approach (S6 is hardly better than S2). This should be discussed.

It would be informative to also show 1:1-scatterplots (AGB predictions vs. observations) for the different methods. And $R^2$ values should be provided, because the given measures of error (RMSE, MRE, MAE) are uninformative with regard to whether there is any trend between predictions and observations at all. I strongly advise to provide $R^2$ values of predictions vs. observations for methods S1-S7.

L. 343: The text in Fig. 6 is too small to read. Also the maps are too small for meaningful interpretation. I suggest to show only map (a) in large and to put map (b) to the appendix. The caption text "green outside of study area" is confusing. Green is part of the DTM color palette.

L. 348-350: The total difference between the county-wide predictions of the optimal model and predictions using the simple allometric approach is only 0.17%. Does this mean that the methodology presented in this study is not needed?

Discussion:

L. 418: This section does not answer the questions raised in the comment on L. 308.

L. 435-469: Comparing absolute RMSE values from different studies representing very different forest types is not meaningful, because the intrinsic AGB variability is very different between, e.g., a tropical rainforest (large) and a Eucalyptus plantation (small). If anything, normalized nRMSE (i.e., RMSE divided by the mean) should be compared.

L. 477: How were the MRE values 0.04% to 99.8% and the average 19.93% calculated? This should be reported in the Materials & Methods and in the Results sections. It could also be visualized in a 1:1-plot. A fundamental question arising from this comparison is, how can we know whether RF&P-BSHADE or the allometric approach is closer to the truth? This would require independent ground-truth plots in other parts of the study area. In fact, an answer to research question (2) raised in the Introduction

would require independent ground-truth plots. Without them I see no justification for the claim that the presented methodology can improve the accuracy of AGB mapping in regions where only non-representative sample units are available (stated in L. 27).

The Discussion and Conclusion are in general very broad and should be more specific with regard to the results in this study.

Supplements:

L. 55-69: Please shorten strongly by removing all information not relevant for the study.

L. 70-115: Please shorten strongly (see comment on L. 192-208 in main text). Please don't repeat text from the Materials & Methods section. Equations are confusing if they are not explained in full detail. E.g., what are w and T in (A.1) etc. In L. 103 a T is mentioned which does not appear in the equations above. In conclusion, I recommend avoiding all equations and explain the mechanics of the different ML methods with words.

L. 116-171: The explanation of P-BSHADE is very technical and non-intuitive. The overview should contain a simple explanation of what it does and how it compares to kriging or IDW. The spatial aspect is not clear at all. What role do plot positions and distances play?

L. 120: What does temperature have to do with it?

L. 188: Table B.4: What is variable coefficient? Do you mean coefficient of variation (CV)? Was longitude used with two decimal precision only? Given the narrow range 117.45 to 117.5, the precision should probably be more than two decimals.

L 197: Fig. C.1: Please increase font size. Labels are missing at the x- and y-axis.

---

## Referee Comment (RC2) · Anonymous Referee #2 · 4 Jul 2019

General comments

This is an excellent study providing most accurate AGB maps based on fusion of forest inventory data and machine learning techniques. The approach combines the advantages of machine-learning algorithms and accounts for spatial correlation among data points, as well as, non-linear relationships between environmental covariates by using the P-BSHADE model. Research questions investigating (1) differences among the presented methods, and (2) how to improve the accuracy of AGB maps are evaluated by providing statistical error metrics, such as mean absolute error (MAE), mean relative error (MRE) and root means squarer error (RMSE). Based on evaluation of these metrics results suggest that among the different methods random forest (RF) in combination with the spatially explicit model produce the highest prediction accuracy for AGB maps and show that in comparison to traditionally applied allometric models estimates are congruent, but differ in local spatial distribution of AGB. Hence, these results indicate that the proposed method based on non-representative sample prediction of AGB maps should be capable of accounting for spatial heterogeneity of AGB and thus could enhance prediction accuracy of AGB maps.

Specific comments

Although this study introduces a very promising methodology – which due to a combination of machine-learning methods with spatial explicit statistical models, should be capable of resolving problems, such as non-linearity, complexity and spatial heterogeneity commonly comprised in available datasets based due to non-representative sampling of forest inventory plots – there are some minor issues that could be addressed:

(1) Given the fact that environmental parameters (i.e. meteorological variables are missing from the analysis the suitability of the proposed technique for forecasting AGB under future climate scenarios cannot be evaluated.

(2) Therefore, this does not allow to infer conclusions about the behavior of non-stationary systems, such as the response of global forest AGB to climatic signals.

(3) However, under steady-state assumptions, the presented approach can be used to derive management plans based on more accurate assessment of AGB from non-representative sampling plots, which can be compared among different geographic regions.

Nevertheless, after accounting for these minor issues this study should represent a valuable asset to the available literature focusing on improving prediction accuracy of

currently available AGB maps.

L41: design(s).

L86: please explain "stability of the second steps".

L94: remove "tantamount" and focus on concise description of the research questions to be investigated in the discussion section, i.e. by (1) comparing the RMSE among different methods and (2) to interpret the accuracy of AGB maps.

L439-453: maybe start of this section based on the advantage of your method over the other studies presented here?

L479-486: this seems to be your main results, put up front and discuss according to the points presented here!

L519 (and throughout the text): please explain "single analytic trees" and "forest resource inventory data".

---

## Author Comment (AC1) · 13 Aug 2019

Dear Editor, Thank you very much for your letter dated 24th July. We are very grateful for the recommendations and comments we have received from the reviewers. We have addressed these comments in the revised manuscript to the best of our ability. All changes made to the original version are marked with RED in revised version, and are listed in the response to the reviewer's comments. Here we attached our response, revised manuscript and Supplemental Material. We believe that the manuscript is now significantly improved and we look forward to your decision. We would also like

to thank the two anonymous reviewers for their constructive comments, which have been very helpful for improving this manuscript. Thank you very much for your kind consideration. Yin Ren

Please also note the supplement to this comment:
https://www.biogeosciences-discuss.net/bg-2019-202/bg-2019-202-AC1-supplement.zip